# Association between Circulating Fibroblast Growth Factor 21 and Aggressiveness in Thyroid Cancer

**DOI:** 10.3390/cancers11081154

**Published:** 2019-08-12

**Authors:** Yea Eun Kang, Jung Tae Kim, Mi Ae Lim, Chan Oh, Lihua Liu, Seung-Nam Jung, Ho-Ryun Won, Kyungmin Lee, Jae Won Chang, Hyon-Seung Yi, Hyun Jin Kim, Bon Jeong Ku, Minho Shong, Bon Seok Koo

**Affiliations:** 1Department of Endocrinology and Metabolism, Chungnam National University College of Medicine, Daejeon 35015, Korea; 2Department of Medical Science, College of Medicine, Chungnam National University, Daejeon 35015, Korea; 3Department of Otolaryngology-Head and Neck Surgery, Chungnam National University College of Medicine, Daejeon 35015, Korea

**Keywords:** FGF21, thyroid cancer, cell proliferation, metastasis, obesity, FGFR

## Abstract

Fibroblast growth factor 21 (FGF21) plays important roles in regulating glucose, lipid, and energy metabolism; however, its effects in tumors remain poorly understood. To understand the role of FGF21 in regulating tumor aggressiveness in thyroid cancer, serum levels of FGF21 were measured in healthy subjects and patients with papillary thyroid cancer (PTC), and expression levels of FGF21, FGF receptors (FGFRs), and β-klotho (KLB) were investigated in human thyroid tissues. The cell viability, migrating cells, and invading cells were measured in PTC cells after treatment with recombinant FGF21. Higher serum levels of FGF21 were found in patients with thyroid cancer than in control participants, and were significantly associated with body mass index (BMI), fasting glucose levels, triglyceride levels, tumor stage, lymphovascular invasion, and recurrence. Serum FGF21 levels were positively correlated with the BMI in patients with PTC, and significantly associated with recurrence. Recombinant FGF21 led to tumor aggressiveness via activation of the FGFR signaling axis and epithelial-to-mesenchymal transition (EMT) signaling in PTC cells, and AZD4547, an FGFR tyrosine kinase inhibitor, attenuated the effects of FGF21. Hence, FGF21 may be a new biomarker for predicting tumor progression, and targeting FGFR may be a novel therapy for the treatment of obese patients with PTC.

## 1. Introduction

The fibroblast growth factor (FGF) ligand family contains 22 known components that can be characterized as hormone-like FGFs (FGF19, FGF21, and FGF23) and canonical FGFs [1,2]. Upregulated expression of various canonical FGFs (including FGF1, FGF2, and FGF 6–9) derived from tumor cells or stromal cells induces tumor progression in various cancers [3,4,5,6]; however, the effects of circulating hormone-like FGFs (such as FGF19, FGF21, and FGF23) on tumors are unclear. Unlike canonical FGFs, FGF21 is a non-classic FGF that lacks affinity for heparin binding and requires β-klotho (KLB) member proteins as co-receptors to activate FGF receptors (FGFRs) [7]. FGF21 plays important roles in the regulation of glucose, lipid, and energy metabolism [7,8,9] and a specific role in adipose tissues via the FGFR1–KLB complex [10,11,12,13]. The liver appears to be the major source of FGF21 in the regulation of metabolic homeostasis in response to various nutritional conditions including fasting, consuming a ketogenic diet, steatosis, and obesity [8,14,15,16].

In response to various stresses, FGF21 is also induced in extrahepatic tissues, such as adipose tissue, pancreas, and skeletal muscle [17,18,19]. Its upregulation also depends on mitochondrial dysfunction and endoplasmic reticulum (ER) stress [20,21,22,23]. Several studies have indicated that circulating FGF21 levels may be a biomarker for predicting the progression of human diseases, including end-stage renal disease, chronic liver disease, and heart failure [24,25,26,27]. FGF21 promotes angiogenesis in vivo and in vitro in extrahepatic tissue through FGFR1 signal induction, and its activation requires internalization of FGFR1 and the KLB co-receptors [28]. However, the exact role of FGF21 in tumor aggressiveness in extrahepatic tissue remains poorly understood.

In many cancers, aberrant deregulation of FGFR signaling is related to tumorigenesis by the upregulation of MAPK signaling or phosphatidylinositol 3-kinase (PI3K) axis [29,30]. Targeting the FGFR signaling axis in thyroid cancer is important for managing advanced and refractory disease, because upregulation of FGFR signaling induces not only tumor progression, but also resistance to anticancer therapy [1,31]. However, the exact mechanism of induction of upregulated FGFR signaling in refractory thyroid cancer remains unclear. 

This study investigated whether FGF21 has a role in regulating tumor aggressiveness in thyroid cancer. We examined expression levels of FGF21, FGFRs, and KLB in human thyroid tissue and the FGF21–FGFR signaling axis in thyroid cancer cells treated with recombinant FGF21 (rFGF21). Additionally, we analyzed the association between serum levels of FGF21 and diverse clinicopathologic parameters in patients with papillary thyroid carcinoma (PTC).

## 2. Results

### 2.1. Clinical Features of PTC Patients

To evaluate the role of FGF21 in human thyroid cancer, serum levels of FGF21 were measured in patients with PTC and control subjects. We enrolled 231 participants from the outpatient clinic of the Division of Otolaryngology Head and Neck Surgery and the Division of Endocrinology and Metabolism of Chungnam National University Hospital in Daejeon, Korea, from January 2008 to July 2017. Inclusion criteria and exclusion criteria are described in the Materials and Methods section. In total, this study included 127 participants with PTC and 52 control subjects matched for age and sex (Figure 1A). Appendix A shows the clinicopathological characteristics of the 127 patients treated for PTC: 85% of the patients were female (109/127), 26.8% had tumors characterized by multicentricity (34/127), 67.7% showed capsular invasion (86/127), 54.3% had extrathyroidal extension (69/127), and 78.7% had lymphovascular invasion (100/127). Of the 127 patients with PTC, 33.1% (42/127) had lymph node (LN) metastases and 10.2% (13/127) had lateral neck LN metastases. Recurrence was observed in 7.1% (9/127) of the patients, the survival rate was 96.9% (123/127), and the mean follow-up period was 84.7 ± 37.4 months. Serum levels of FGF21 were measured in all participants using an ELISA.

### 2.2. Higher Levels of FGF21 in Patients with PTC

FGF21 levels were significantly higher in the PTC group than in controls (Figure 1B). No significant difference between the groups was detected in the majority of the metabolic parameters, including body mass index (BMI), body weight, fasting glucose, triglycerides (TG), total cholesterol (TC), high-density lipoprotein cholesterol (HDL-C), and low-density lipoprotein cholesterol (LDL-C) (Table 1).

### 2.3. Relationship between FGF21 Levels and Clinicopathological Parameters

We analyzed the relationships between clinicopathological parameters and serum levels of FGF21 in patients with PTC. The median FGF21 level was 184.1 pg/mL in patients with PTC; the patients were divided into high FGF21 and low FGF21 groups based on the median value (Table 2). Univariate analyses revealed that FGF21 levels were significantly associated with several clinicopathological parameters including T stage (*p* = 0.001), microscopic capsular invasion (*p* = 0.011), extrathyroidal extension (*p* = 0.003), locoregional recurrence (*p* = 0.017), and survival (*p* = 0.044) (Table 2). FGF21 is induced by metabolic stress caused by conditions such as obesity, non-alcoholic fatty liver disease, and diabetes mellitus, as well as by damage to various organs, which involves mitochondrial and ER stress. To investigate the increased serum levels of FGF21 in patients with aggressive tumors, we compared various metabolic parameters between groups with high and low FGF21 levels (Table 2). The high FGF21 group had significantly higher BMIs, fasting glucose levels, and fasting TG levels. Additionally, we used univariate analysis to analyze clinicopathological parameters according to serum FGF21 levels. Correlation analyses revealed that FGF21 levels were weakly correlated with BMI (Appendix A and Figure 1C), significantly associated with serum TG levels, and negatively correlated with serum HDL levels (Appendix A).

### 2.4. Multivariate Analysis of the Relationships between FGF21 Levels and Tumor Aggressiveness

To determine the role of FGF21 as an independent determinant of aggressive phenotypes in PTC, we conducted multivariate analyses using stepwise logistic regression to further examine the parameters that were significant in univariate analyses (Table 3). Multivariate analyses revealed that high FGF21 level was an independent risk factor for recurrence (odds ratio, 9.985; *p* = 0.038).

### 2.5. Correlations between FGF21 Levels and Recurrence-Free Survival and Overall Survival in Patients with PTC

Kaplan–Meier survival analyses revealed that the five-year recurrence-free survival rate was 85.9% in the high FGF21 group and 98.4% in the low FGF21 group. Patients with high FGF21 levels had a significantly lower rate of five-year recurrence-free survival than patients with low FGF21 levels (Figure 2A). In addition, the overall survival rate differed significantly between the two groups, being 93.8% in the high FGF21 group and 100% in the low FGF21 group (Figure 2B). These data suggested that the serum level of FGF21 is a new predictive biomarker for recurrence and mortality in patients with PTC.

### 2.6. FGF21 Promotes Migration and Invasion of Thyroid Cancer Cell Lines

FGF21 helps regulate homeostasis by binding to FGFRs and the KLB cofactor; the expression levels of FGF21, KLB, and FGFRs were investigated in this study. We hypothesized that the expression of FGF21 may be induced in thyroid tissues in response to thyroid tumorigenesis, because FGF21 is induced in extrahepatic tissues under high stress conditions, although FGF21 originates mainly from the liver [17,18,19]. Previous studies about the expression of FGFRs in thyroid cancer identified the upregulation of FGFR1 or FGFR4 in thyroid cancers, compared to normal tissues [32,33]. However, there is no study about the expressions for both FGFRs and KLB in human samples using TCGA database or Western blot analysis. We first compared the protein expression levels of FGF21 and FGFRs between normal and tumor tissues (Figure 3A). Although both FGFRs and KLB were expressed in both normal thyroid and tumor tissues, FGF21 expression was not detected using Western blot analysis. We identified the significant upregulation of FGFR4 expression in tumors, compared to that in normal tissues, whereas other KLB, FGFR1, FGFR2, FGFR3, and phospho-FGFR revealed no significant differences between tumors and normal tissues. Next, we examined the expression levels of FGF21, FGFRs, and KLB using The Cancer Genome Atlas (TCGA) database (Figure 3B). TCGA database revealed that all FGFRs and KLB were expressed in both normal tissues and tumors; however, expression of FGF21 was very low in most samples, and only that of FGFR3 was significantly higher in tumors compared to normal samples. The upregulation of FGFR4 in our tumor samples compared to normal tissues in Western blot supports previous results of the other study [32]. However, the inconsistent findings of FGFRs between protein expression and mRNA expression from TCGA database in the present study suggested that further studies are needed to elucidate the FGFR signaling in thyroid cancers.

To determine the effect of FGF21 on tumor aggressiveness in thyroid cancer, we evaluated the expression levels of FGFRs, such as FGFRs and KLB, and phospho-FGFR in normal thyroids and thyroid cancer cell lines. Western blot analyses revealed that all receptors for FGF21 were expressed in all cancer cells and normal cells (Figure 3C and Appendix A).

We focused on the endocrine effects of FGF21 in the tumorigenesis of differentiated thyroid carcinomas using PTC cell lines (TPC-1 and BCPAP) treated with rFGF21. However, WST-1 assays revealed that the viability of PTC cells was not significantly affected by rFGF21 (Figure 3D,E). In addition, we evaluated the effect of rFGF21 on Nthy-ori3-1, which is a normal thyroid follicular epithelial cell line (Appendix A). The viability of Nthy-ori3-1 was also not significantly affected by rFGF21. To evaluate the role of FGF21 in tumor aggressiveness, we next investigated the effects of rFGF21 treatment on the migration and invasion of thyroid tumor cells (Figure 3F–I). The number of migrated cells was significantly increased in FGF21-treated cells during 24 h incubation of TPC-1 and BCPAP cells (132.5% and 125.5%, respectively) compared to controls. Moreover, the number of invading cells significantly increased after 24 h incubation of TPC-1 and BCPAP cells with FGF21 (265% and 370.5%, respectively) compared to controls. The number of migrated and invading cells also increased after 3, 6, and 12 h incubation with FGF21 (Appendix A). In a normal thyroid cell line, Nthy-ori3-1, there was no significant effect of FGF21 on migration (Appendix A).

### 2.7. FGF21 Activates Epithelial–Mesenchymal Transition (EMT) Signaling and Upregulates FGFR Signaling

Treatment with rFGF21 induced a significant increase in the migration and invasion of thyroid cancer cells. The levels of vimentin, a molecule involved in EMT signaling, were significantly higher in rFGF21-treated BCPAP cells than in controls (Figure 4A and Appendix A). In addition, the levels of E-cadherin were lower in rFGF21-treated TPC-1 cells than in controls (Figure 4B and Appendix A). To determine the mechanism responsible for the promotion of aggressiveness of thyroid cancer cells by FGF21, we analyzed the downstream targets of FGFR pathways. In rFGF21-treated thyroid cancer cells, phosphorylation of FGFR was increased. Phosphorylation of AKT was also increased, whereas expression of total AKT did not change. Moreover, both BCPAP and TPC-1 cells showed increased phosphorylation of ERK, whereas the total ERK levels did not change (Figure 4 and Appendix A). These results suggested that rFGF21 treatment promoted tumor aggressiveness by upregulating the FGFR signaling axis through phosphorylation of AKT and ERK in differentiated thyroid carcinoma cells.

### 2.8. AZD4547 Attenuates the Effects of rFGF21 in Thyroid Cancer via Downregulation of FGFR Signaling

Our clinical results revealed that increased serum levels of FGF21 in PTC patients were significantly associated with poor prognosis. To determine whether targeting FGFR had an effect on PTC cells treated with rFGF21, we examined the effects of treatment with AZD4547, a FGFR tyrosine kinase inhibitor, after rFGF21 treatment (Figure 5 and Appendix A).

We compared cell migration and invasion between cells with and without AZD4547 treatment after rFGF21 administration. The results showed that AZD4547 treatment resulted in a significant decrease in migrated and invading cells, compared to the control group (Figure 5A–D and Appendix A). Because our results indicated that rFGF21 upregulated EMT signaling via the FGFR signaling axis in PTC cells, we conducted Western blot analyses to examine the effects of AZD4547 (Figure 5). The results revealed that AZD4547 treatment decreased EMT signaling and the FGFR signaling axis, including phosphorylation of AKT and ERK (Figure 5E,F and Appendix A). Our results suggested that the serum levels of FGF21 may be related to obesity or hyperglycemia in patients with PTC, and that FGF21 promotes tumor progression via upregulation of the FGFR signaling axis, including phosphorylation of AKT and ERK. Notably, targeting FGFR attenuated the effects of FGF21 on PTC cells (Figure 5G).

## 3. Discussion

We demonstrated a novel effect of FGF21 in the regulation of tumor aggressiveness in thyroid cancer. We investigated whether the involvement of FGF21 in tumor progression occurred via upregulation of EMT signaling and FGFR signaling. Our findings indicated that high serum levels of FGF21 were significantly associated with recurrence-free survival and overall survival of patients with PTC, which suggested that FGF21 could serve as a new biomarker for predicting tumor progression. Additionally, serum levels of FGF21 were associated with metabolic parameters, including BMI, serum glucose levels, and serum TG levels, suggesting that FGF21 may mediate the association between obesity and tumor progression in PTC.

Abundant FGF21 expression is induced in the liver under normal physiological conditions [34]; moreover, it is expressed in other tissues, such as the pancreas, adipose tissue, muscle, and kidney under pathological stress conditions [19,20,35]. FGF21 is a well-known hormone-like FGF that significantly increases in expression in response to various nutritional and metabolic perturbations, and alleviates metabolic stress [20,34]. Previous studies reported that a methionine- and choline-deficient lean diet, and hepatocarcinogenesis, induced FGF21 expression in a manner independent of obesity and diabetic conditions [20,36,37]. FGF21 expression in the pancreas has a protective role against proteotoxic and ER stress [38,39], and FGF21 is significantly induced in muscles undergoing hypertrophy, or showing mitochondrial deficiency or metabolic abnormalities [40,41]. These findings suggest that FGF21 is an important stress-inducible cytokine in most tissues, and may have tissue-specific functions. Previously, we described a reduction of oxidative phosphorylation in patients with thyroid cancer harboring the BRAF mutation [42], and we hypothesized that FGF21 was induced in the thyroid due to mitochondrial stress associated with thyroid tumorigenesis [43]. However, thyroid cancer cells treated with doxycycline, an inducer of mitochondrial stress, did not induce FGF21 expression. Thus, in the present report, we investigated the serum levels of FGF21 in patients with thyroid cancer, and examined various metabolic parameters in patients categorized into high and low FGF21 groups.

Notably, BMI, fasting glucose levels, and TG levels were significantly higher in the high FGF21 group. Correlation analyses revealed that FGF21 levels were related to BMI. These results were similar to those of previous reports that focused on the effects of obesity on tumor aggressiveness in thyroid cancer [44,45]. Obese patients with thyroid cancer show a very aggressive phenotype in terms of tumor size, tumor stage, and LN metastasis, which suggests that the factors mediating the effects of obesity on tumor progression are thyroid hormones, insulin resistance, and adipokines [44,45]; however, no previous studies have investigated the relationship between serum levels of FGF21 and tumor aggressiveness. Our study demonstrated that upregulation of FGF21 due to metabolic stress had an important role in tumor progression in thyroid cancer. The findings indicated that upregulation of circulating FGF21 was linked with adiposity or glucose intolerance in patients with thyroid cancer, whereas previous studies focused on the role of FGF21 in metabolic diseases. Although our results suggested that metabolic dysregulation increased serum FGF21 levels, we were unable to precisely determine the origin of FGF21 in thyroid cancer patients. We hypothesized that the liver might be the major organ producing FGF21, because serum FGF21 is secreted primarily by the liver during overfeeding to regulate metabolism [46]. We believe that further studies focusing on the exact mechanism of transcriptional regulation of FGF21 in systemic diseases are therefore necessary.

Several studies have reported increased serum levels of FGF21 in various human cancers, including clear cell renal cell carcinoma and breast cancer [47,48]. Although previous studies have suggested that FGF21 levels are a promising predictive biomarker for tumor progression, no previous study has identified the mechanism underlying the effects of FGF21 on tumor progression in extrahepatic tissues. In our study, treatment of thyroid cancer cell lines with rFGF21 induced tumor progression, including migration and invasion, along with upregulation of the EMT signaling axis, whereas there was no significant increase in the viability of tumor cells in thyroid cancer. The phosphorylation of ERK and AKT, which are downstream targets of the FGFR signaling axis, was upregulated after rFGF21 treatment in thyroid cancer cells. Additionally, elevated serum FGF21 was significantly related to aggressive tumor phenotypes. Taken together, FGF21 promoted migration and invasion of thyroid cancer cells by upregulating FGFR signaling.

Our results are inconsistent with previous studies regarding the beneficial effects of FGF21 against metabolic disease. Recent studies have indicated specificity of FGF21 for FGFR1-KLB in adipose tissue [11,12,13], and that the administration of rFGF21 to obese and diabetic patients does not stimulate tumor growth [10,49,50]. Previous studies have suggested that the lack of proliferation in the presence of FGF21 may be due to the distinct intracellular environment in KLB-expressing cells, or that the action of KLB in specific cell components may be related to inhibition of the cell proliferation pathway. As in previous studies, we also investigated the expression of KLB, a major FGFR. Our results revealed that protein expression of KLB was detected in thyroid tissue. A previous study about the expression of FGFRs in thyroid tissues using Western blot analysis, as used in our experiment, identified that not other FGFRs but only FGFR2 was expressed in human normal thyroid tissue [32]. They identified upregulation of FGFR4 in aggressive thyroid cancer, compared to normal thyroid tissue using immunohistochemical analysis [32]. Another study also identified the upregulation of FGFR1 expression in human thyroid cancer, compared to normal [33]. Interestingly, a recent study focused on the role of FGF19/FGFR4/KLB signaling pathway in the development of thyroid cancers with upregulation of serum levels of KLB, FGF19, and FGFR4 in patients with thyroid cancer, compared to healthy controls [51]. The upregulation of protein expression of FGFR4 in tumors compared to normal tissues in the present study supports the previous results of another study [32], but we could not find the significant difference of other receptors for FGF21, including KLB and FGFR1-3, between tumors and paired normal tissues. Additionally, results from TCGA database were not consistent with previous results. These findings suggested that further studies are needed to elucidate the FGFR signaling, including the role of KLB in thyroid cancers.

Our results are similar to those of a previous report describing the role of FGF19 (another hormone-like FGF similar to FGF21) in a cancer model. That report showed that FGF19 increased the phosphorylation of GSK-3 beta and activated the beta-catenin signaling axis, mediated by the regulation of FGFR4 expression. Additionally, other studies have suggested that FGFR4 knockdown leads to slower tumor progression in a rodent model of colon cancer [52,53]. Moreover, although FGF19 has minimal mitogenic activity in fibroblasts in vitro, the skeletal muscles of FGF19 transgenic mice develop hepatocellular carcinoma in a manner dependent on FGFR4 expression [53,54]. The effects of FGF21 in our study may have been induced by the upregulation of FGFR signaling, which suggests that FGF21 may induce growth-stimulating effects in tumor cells in a manner dependent on the expression of FGFRs and KLB.

Previous studies reported that FGF21 overexpression increased life span through downregulation of the hepatic expression of IGF-1 by inhibition of the STAT5 transcription factor [55]. Other studies focused on the role of FGF21 as the major mediator of the association between low cancer risk and a vegan diet, via downregulation of serum IGF-1 levels [56]. These studies showed that FGF21 may prevent cancer, and promote leanness and insulin sensitivity. However, these effects were only observed in mouse models featuring highly overexpressed FGF21, which represents a metabolic healthy phenotype. Therefore, these canonical effects of FGF21 may not be directly relevant to the role of FGF21 in human cancer, and there has been no study describing the specific role of FGF21 in tumor cells. Similar to our results, another study provided evidence that serum FGF21 levels mediate the association between metabolic syndrome and colorectal cancer risk [57]. Aberrant regulation of the FGF signaling axis is involved in the pathogenesis of malignancies via upregulation of the MAPK cascade or PI3K/AKT signaling axis [1,3]. In addition, as upregulation of FGF signaling induces not only tumor progression, but also resistance to anticancer therapies in many tumor types [1,58,59], multiple studies have proposed that aberrant FGF signaling could be a therapeutic target for various tumors, including thyroid cancer [1,3,58]. The FGFR signaling axis in thyroid cancer is important for the management of advanced and refractory disease [31,60]. Treatment with lenvatinib, a multikinase inhibitor targeting receptors for vascular endothelial growth factor, FGF, and platelet-derived growth factor, improves progression-free survival in patients with advanced thyroid cancer and iodine-refractory differentiated thyroid cancer [31], and shows anti-angiogenic effects involving inhibition of FGFR signaling [61,62]. However, no previous studies have reported the exact mechanism underlying the upregulation of FGFR signaling in human thyroid cancer. Our study revealed that FGFR signaling was upregulated in thyroid cancer in response to increased serum levels of FGF21 due to metabolic stress. In addition, targeting FGFR attenuated the effects of FGF21 on PTC cells.

Our study had several limitations. We only measured serum levels of FGF21 in patients with thyroid cancer at the time of initial diagnosis. To validate our concept of serum FGF21 level as a predictive biomarker in thyroid cancer, further long-term follow-up studies that measure the serum levels of FGF21 in patients with thyroid cancer at the initial diagnosis, after thyroidectomy, and in those with stable and recurred disease status are necessary. Another important limitation was that we could not fully determine the direct biological effects of FGF21 due to the discrepancy in FGF21 concentrations between the in vitro assays and human data, as well as the lack of a precise method for measuring active forms of circulating FGF21. In spite of these limitations, our study is the first in humans to identify the important mediating role of FGF21 in the relationship between thyroid cancer aggressiveness and metabolic dysregulation, and suggested that increased serum levels of FGF21 in patients with thyroid cancer may be a useful biomarker for tumor progression.

## 4. Materials and Methods

### 4.1. Subjects

The inclusion criteria were as follows: age >18 years, absence of any clinical sign of infection or inflammation, no alcohol or drug abuse, systolic blood pressure <140 mm Hg and diastolic blood pressure <100 mm Hg, no pregnancy, and no previous history of diabetes mellitus. A total of 18 participants were excluded because of liver cirrhosis (3 cases), other malignancies (1 case of prostate cancer, 2 cases of liver cancer, and 1 case of pituitary adenoma), and type 2 diabetes mellitus (11 cases). In total, this study included 127 participants with PTC and 40 control participants. In our study, all participants were diagnosed with PTC by ultrasound-guided fine needle aspiration biopsy. Among the 127 participants, 12 diagnosed with papillary microcarcinoma without extrathyroidal extension and LN metastasis underwent lobectomy, but not central LN dissection; 80 underwent total thyroidectomy with prophylactic central LN dissection without clinical evidence of LN on imaging or palpation; 32 with clinically evident positive central LN underwent total thyroidectomy with therapeutic central LN dissection; and the remaining 3 underwent total thyroidectomy with central and lateral LN dissection due to evidence of metastatic LNs in the lateral neck before surgery. Lateral LN dissection was performed using a modified radical operation that involved complete removal of level II–V lateral cervical LNs. Level I dissections were not performed in patients lacking clinical evidence of metastases at level I. All specimens were collected from patients after informed consent was obtained in accordance with the Institutional Guidelines of Chungnam National University Hospital. Diagnoses were made according to the World Health Organization (WHO) classification of endocrine organ tumors [63]. Typically, recurrent disease was checked for via physical examinations, measurement of serum levels of thyroglobulin and anti-thyroglobulin, and ultrasonography (every 12 months for 5 years). For patients suspected of disease recurrence, we conducted fine needle aspiration biopsy, whole body scan, and computed tomography of the neck. The following criteria were used to define recurrence—either pathological evidence of disease on excision or cytology, or recurrent disease confirmed by elevated Tg and scanning studies. Patients with no recurrence after a 60-month follow-up, as determined from clinical, laboratory, and radiological (neck ultrasonography) studies, were categorized into the nonrecurrent group. The median follow-up period to evaluate tumor recurrence was 84.7 ± 37.4 months. The protocol for this study was approved by the Institutional Review Board of Chungnam National University Hospital (Reg. No. CNUH IRB No. 2017-07-005).

### 4.2. Clinical and Biochemical Parameters

All participants underwent physical examinations on the first day of the study. Height, body weight, systolic blood pressure, and diastolic blood pressure were recorded. The blood pressure of each patient was measured after 10 min of rest via the right arm in the seated position using an Omron IntelliSense Automatic Blood Pressure Monitor (Omron, Kyoto, Japan). The height and body weight were measured without shoes in the morning. The BMI was calculated as weight in kg divided by height in m^2^. All blood samples were collected into tubes containing EDTA in the morning, after an overnight fast of more than 8 h. We measured the levels of fasting glucose, TG, (TC), LDL-C, HDL-C, aspartate aminotransferase, alanine aminotransferase, and thyroid stimulating hormone (TSH). Blood chemistry and lipid profiles were assessed using a blood chemistry analyzer (747; Hitachi, Tokyo, Japan).

### 4.3. Serum Levels of FGF21

Blood samples for measurements of FGF21 were collected at the time of the initial diagnosis of thyroid cancer. Venous blood samples were acquired after a fast of at least 8 h on the morning of thyroid surgery. In control participants, blood samples were collected using the same method as for the thyroid cancer patients. Plasma was prepared by centrifugation at 3000 rpm at 4 °C for 15 min, and was then stored in liquid nitrogen until analysis. Serum levels of FGF21 were measured using a quantitative sandwich ELISA kit (Catalog No. DF2100; R&D Systems, Minneapolis, MN, USA).

### 4.4. Genomic and Clinical Data Sets

All genomic data of papillary thyroid carcinoma from TCGA project were obtained from TCGA data portal [64] and cancer browser [65]. A heatmap was generated using Cluster and TreeView software programs (version 1.6, Glasgow University, Glasgow, UK).

### 4.5. Cell Lines and Cell Cultures

Nthy-ori-3-1, a normal thyroid follicular cell line from a human adult, was obtained from Dr. Anna Maria Porcelli (Bologna University, Bologna, Italy). The human PTC cell lines, BCPAP and TPC-1, were provided by Dr. M. Santoro (Università di Napoli Federico II, Naples, Italy) and Dr. Masahide Takahashi (Nagoya University, Nagoya, Japan), respectively. BCPAP and TPC-1 were maintained in Dulbecco’s Modified Eagle’s Medium (DMEM; Invitrogen, Carlsbad, CA, USA). Nthy-ori3-1 and 8505c cells were cultured in RPMI 1640 (Invitrogen). Both types of media were supplemented with 10% heat-inactivated fetal bovine serum (FBS; Invitrogen), 100 U/mL penicillin, and 100 g/mL streptomycin (Invitrogen). Cells were cultured at 37 °C in a humidified chamber with a 5% CO_2_ atmosphere. Recombinant FGF21 (R&D Systems) treatment is described in Appendix A and Methods.

### 4.6. Recombinant FGF21 Treatment

Recombinant FGF21 (R&D Systems) was reconstituted at 100 μg/mL in sterile phosphate-buffered saline containing at 0.1% BSA (Invitrogen). After 24 h of serum starvation, PTC cells (BCPAP and TPC-1 cells) and normal thyroid cells (Nthy-ori3-1) were exposed to recombinant FGF21 at different concentrations.

### 4.7. Western Blot Analysis

Human tissues including normal thyroid tissues and thyroid cancer tissues and cells were lysed in RIPA buffer (30 mm Tris, pH 7.5, 150 mm sodium chloride, 1 mm phenylmethylsulfonyl fluoride, 1 mm sodium orthovanadate, 1% Nonidet P-40, and 10% glycerol) containing phosphatase and protease inhibitors (Roche, Basel, Switzerland). Western blot analyses were performed with 30–50 μg protein from the tissue or cell homogenates using commercially available antibodies. Primary antibodies were tested against glyceraldehyde 3-phosphate dehydrogenase (1:2000; Cell Signaling Technology, Beverly, MA, USA), beta-actin (1:2000; Abcam, Cambridge, UK), KLB (1:1000; Life Span Bioscience, Seattle, WA, USA), FGFR1 (1:1000; Invitrogen), FGFR2 (1:1000; Abcam), FGFR3 (1:1000; Abcam), FGFR4 (1:1000; Abcam), AKT (1:1000; Cell Signaling Technology), phospho-AKT (1:1000; Cell Signaling Technology), ERK (1:1000; Cell Signaling Technology), phospho-ERK (1:1000; Cell Signaling Technology), vimentin (1:1000; Santa Cruz Biotechnology, Santa Cruz, CA, USA), N-cadherin (1:1000; Santa Cruz Biotechnology), E-cadherin (1:1000; Santa Cruz Biotechnology), SLUG (1:1000; Cell Signaling Technology), TWIST (1:1000; Cell Signaling Technology), and phospho-FGFR (1:1000; Cell Signaling Technology). Immunoreactive bands were visualized on polyvinylidene difluoride membranes (Thermo Fisher Scientific, Waltham, MA, USA) using alkaline phosphate-linked anti-rabbit (1:2000; Abcam) or anti-mouse antibody (1:2000; Abcam) and an enhanced chemiluminescence detection system (LI-COR Biosciences; Lincoln, NE, USA). The detailed methods are described in the Appendix A and Methods.

### 4.8. Cell Viability Assay

Cells were plated at a density of 5 × 10^3^/well in serum-free culture medium, and treated with 0.1% BSA (control) or rFGF21 (200 ng/mL). After 12 h incubation, the viability of PTC cells was measured using the WST-1 Cell viability reagent (Roche Diagnostics Corporation, Indianapolis, IN, USA) as described previously [66]. The results are presented as the percentage of the control cell value.

### 4.9. Cell Migration and Invasion Assay

Migration of PTC cells was performed in Transwell chambers (Corning Costar, Cambridge, MA, USA) with 6.5 mm diameter polycarbonate filters (8 μm pore size). The lower surface of the filter was coated with 10 µg gelatin for the migration assay and 25 µg reconstituted basement membrane substance (BD Matrigel Matrix, Catalogue number 354234; BD Bioscience, Heidelberg, Germany) for invasion assays. Fresh DMEM containing 3% FBS was placed in the lower wells. PTC cells (BCPAP and TPC-1) were incubated for 24 h in medium containing 1% FBS, trypsinized, and suspended at a final concentration of 1 × 10^6^ cells/mL in medium containing 1% FBS. Cell suspensions (100 μL) were loaded in the upper wells, and the chambers were incubated at 37 °C for 12 h with control (0.1% BSA) or rFGF21 (200 ng/mL). Cells were fixed and stained with crystal violet. Non-migrating cells on the upper surface of the filter were removed by wiping them with a cotton swab. Chemotaxis was quantified by counting the cells that migrated to the lower side of the filter using an optical microscope (200×); eight randomly chosen fields were counted for each migration assay.

### 4.10. Statistics

The significance of between-group differences was evaluated using Student’s *t*-test or the Mann–Whitney U test, and continuous variables are expressed as means ± SD. Between-group differences were compared using the chi-squared test, and categorical variables are expressed as percentages. To evaluate associations between serum levels of FGF21 and other variables, we subjected the data to Pearson and partial correlation analyses. A two-tailed *p*-value of <0.05 was considered statistically significant. All statistical analyses were performed with SPSS version 22.0 (IBM Corp., Armonk, NY, USA).

## 5. Conclusions

In conclusion, FGF21 had a novel effect on tumor progression through upregulation of the EMT signaling axis in differentiated thyroid cancer. Our findings demonstrated the role of serum FGF21 as a predictive biomarker for tumor aggressiveness in thyroid cancer. Serum FGF21 levels were positively correlated with the BMI in patients with PTC, and targeting FGFR may serve as an alternative therapeutic approach for treatment of obese patients with PTC. 

## Figures and Tables

**Figure 1 cancers-11-01154-f001:**
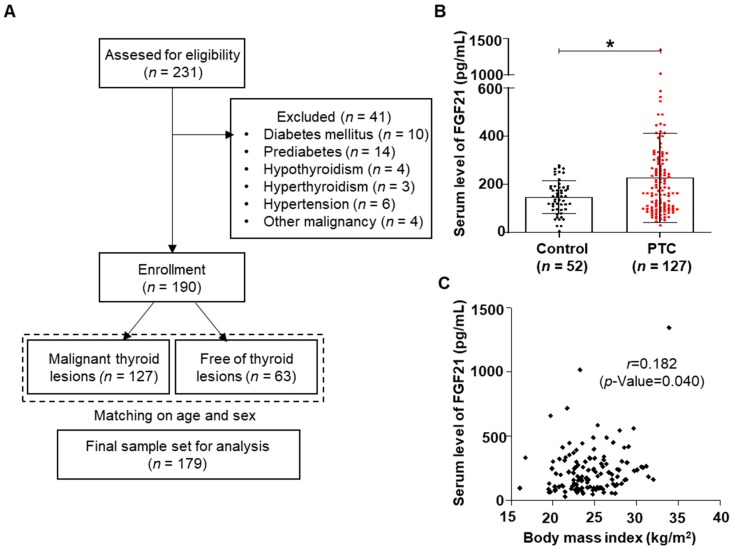
Thyroid cancer and serum levels of fibroblast growth factor 21 (FGF21). (**A**) The overall study design, including the numbers of participants included in this study. (**B**) Serum FGF21 levels in patients with papillary thyroid cancer (PTC) were significantly higher than in control subjects (* *p* < 0.05). (**C**) Serum FGF21 levels in PTC patients correlated positively with the body mass index.

**Figure 2 cancers-11-01154-f002:**
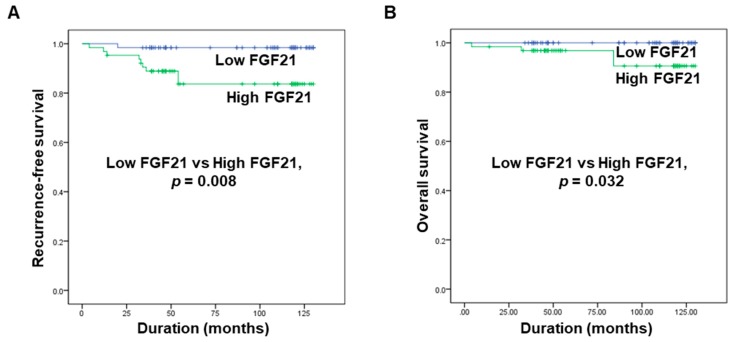
Associations of serum levels of FGF21 with recurrence-free survival and overall survival in patients with PTC. (**A**) Comparison of the five-year recurrence-free survival rate between the high and low FGF21 groups. (**B**) Comparison of overall survival between the high and low FGF21 groups.

**Figure 3 cancers-11-01154-f003:**
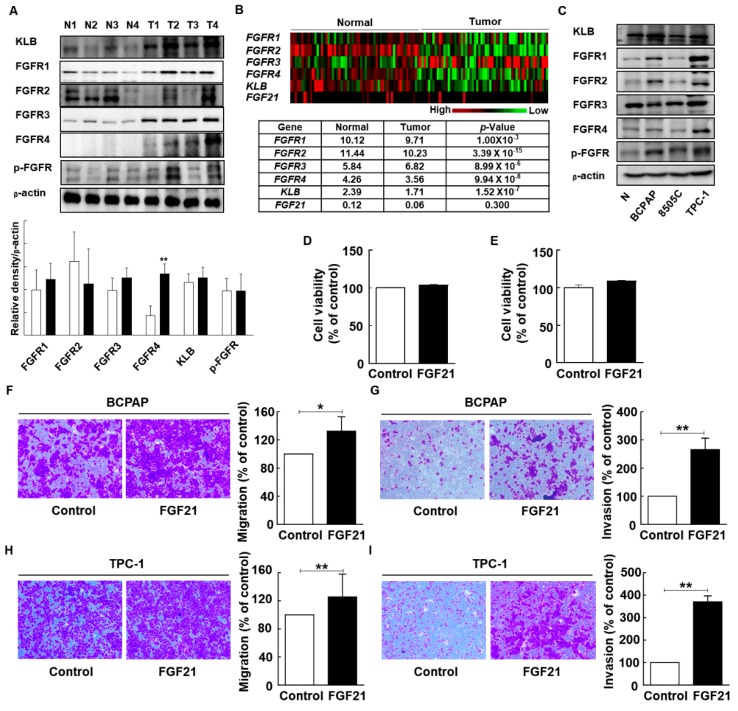
Expression of FGF receptors (*FGFRs*) in human thyroid tissue and the effects of FGF21 on the viability, migration, and invasion of thyroid carcinoma cells. (**A**) Western blot analyses of β-klotho (KLB) and FGFR protein levels in total cell lysates from paired clinical specimens of normal (N) and tumor (T) tissues from four patients with PTC (white bar, normal; black bar, tumor). (**B**) Gene expressions of *FGFR1*, *FGFR2*, *FGFR3*, *FGFR4*, *KLB*, and *FGF21* between normal tissues and paired tumor tissues from a database of 59 patients in TCGA. (Data are given in matrix format, in which rows represent individual genes and columns represent each patient.) (**C**) Protein expression of KLB and all FGFRs in normal thyroid cells (Nthy-ori3-1) and thyroid carcinoma cells (BCPAP, TPC-1, and 8505C). (**D**) The effects of FGF21 on the cell viability of BCPAP cells treated with recombinant FGF21 (rFGF21) or 0.1% bovine serum albumin (BSA; negative control) for 12 h, as determined by WST-1 assay. (**E**) The effects of FGF21 on the cell viability of TPC-1 cells treated with rFGF21 or 0.1% BSA for 12 h, as determined by WST-1 assay. (**F**) The effects of FGF21 on the migration of BCPAP cells treated with rFGF21 or 0.1% BSA for 12 h, as determined by Transwell chamber assay. (**G**) The effects of FGF21 on the invasion of BCPAP cells in chambers coated with Matrigel after treatment with rFGF21 or 0.1% BSA for 12 h. (**H**) The effects of FGF21 on the migration of TPC-1 cells treated with rFGF21 or 0.1% BSA for 12 h, as determined by Transwell chamber assay. (**I**) The effects of FGF21 on the invasion of TPC-1 cells in chambers coated with Matrigel after treatment with rFGF21 or 0.1% BSA for 12 h. The experiments were performed in duplicate, and all experiments were performed at least three times. * *p* < 0.05; ** *p* < 0.01.

**Figure 4 cancers-11-01154-f004:**
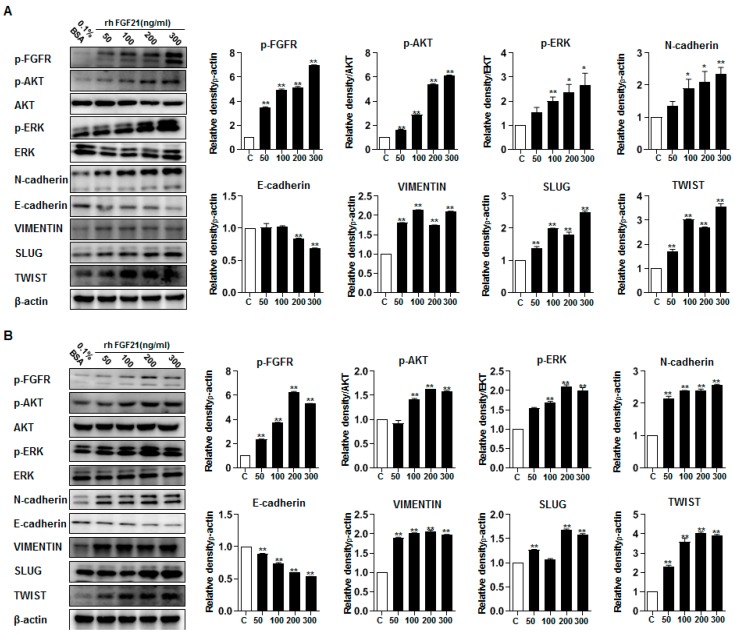
Effects of FGF21 in the FGFR signaling axis, including epithelial–mesenchymal transition (EMT)-associated proteins in thyroid cancer cell lines. (**A**) Representative images of Western blot analyses for the detection of p-FGFR, p-AKT, AKT, p-ERK, ERK, N-cadherin, E-cadherin, vimentin, SLUG, TWIST, and beta-actin in BCPAP cells treated with rFGF21 or 0.1% BSA. (**B**) Representative images of Western blot analyses for the detection of p-FGFR, p-AKT, AKT, p-ERK, ERK, N-cadherin, E-cadherin, vimentin, SLUG, TWIST, and beta-actin in TPC-1 cells treated with rFGF21 or 0.1% BSA. Experiments were performed in duplicate, and all experiments were performed at least three times. ** *p* < 0.01, * *p* < 0.05 compared to the 0.1% BSA control.

**Figure 5 cancers-11-01154-f005:**
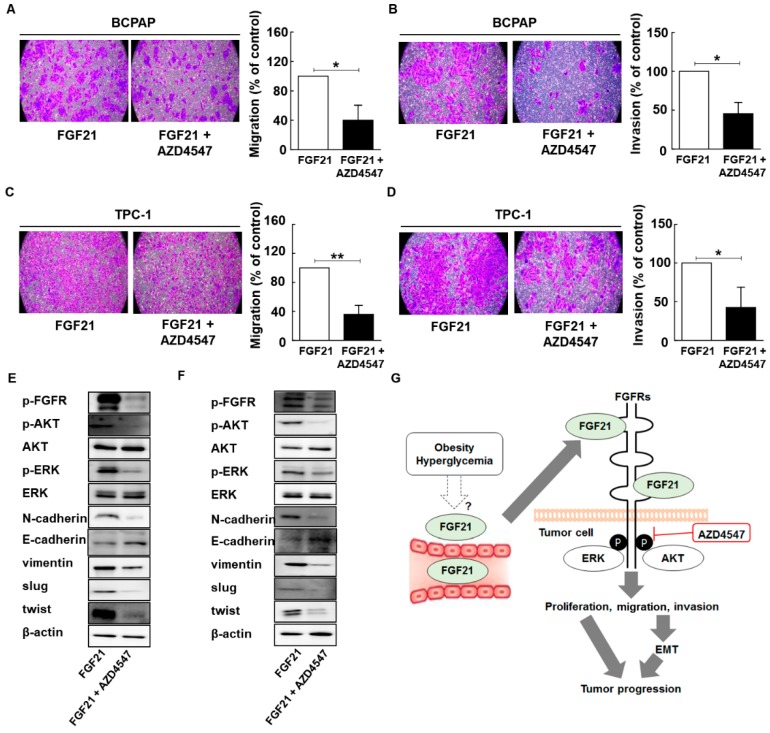
AZD4547 attenuates the effects of FGF21 in thyroid cancer via downregulation of FGFR signaling. (**A**) The effects of AZD45487 on the migration of BCPAP cells after treatment with only rFGF21 or rFGF21 and AZD4547 for 12 h, as determined by Transwell chamber assay. (**B**) The effects of AZD4547 on the invasion in BCPAP cells in chambers coated with Matrigel after treatment with only rFGF21 or rFGF21 and AZD4547 for 12 h. (**C**) The effects of AZD45487 on the migration in TPC-1 cells after treatment with only rFGF21 or rFGF21 and AZD4547 for 12 h, as determined by Transwell chamber assay. (**D**) The effects of AZD4547 on invasion in TPC-1 cells in chambers coated with Matrigel after treatment with only rFGF21 or rFGF21 and AZD4547 for 12 h. (**E**) Representative images of Western blot analyses for the detection of p-FGFR, N-cadherin, E-cadherin, vimentin, p-AKT, AKT, p-ERK, ERK, SLUG, TWIST, and beta-actin in BCPAP cells treated only with rFGF21 or rFGF21 and AZD4547. (**F**) Representative images of Western blot analyses for the detection of p-FGFR, N-cadherin, E-cadherin, vimentin, p-AKT, AKT, p-ERK, ERK, SLUG, TWIST, and beta-actin in TPC-1 cells treated only with rFGF21 or rFGF21 and AZD4547. (**G**) Proposed model of the interaction between serum levels of FGF21 and tumor aggressiveness in PTC based on this study. Experiments were performed in duplicate and all experiments were performed at least three times.

**Table 1 cancers-11-01154-t001:** Comparison of metabolic parameters and serum levels of fibroblast growth factor 21 (FGF21) between the control and papillary thyroid cancer (PTC) groups.

Variable	Control Group (*n* = 52)	PTC Group (*n* = 127)	*p*-Value
Age (years)	52.0 ± 12.3	50.5 ± 12.2	0.488
Sex (male:female)	9:31	18:109	0.212
Body weight (kg)	62.5 ± 11.4	61.2 ± 9.5	0.519
Height (cm)	160.4 ± 9.5	158.0 ± 8.1	0.102
BMI (kg/m^2^)	23.5 ± 2.7	24.5 ± 3.1	0.060
Systolic blood pressure (mmHg)	118.7 ± 9.4	117.7 ± 8.0	0.053
Diastolic blood pressure (mmHg)	74.0 ± 8.9	75.2 ± 5.9	0.318
Fasting glucose (mm)	99.9 ± 9.9	96.4 ± 15.4	0.174
Triglycerides (mm)	137.2 ± 116.1	136.2 ± 83.7	0.961
Total cholesterol (mm)	184.1 ± 32.7	187.9 ± 36.7	0.532
LDL cholesterol (mm)	111.3 ± 29.9	112.1 ± 36.0	0.895
HDL cholesterol (mm)	52.8 ± 13.1	51.6 ± 12.4	0.623
AST (IU/L)	21.1 ± 6.2	19.5 ± 7.2	0.185
ALT (IU/L)	20.3 ± 14.1	20.3 ± 12.4	0.895
TSH (IU/mL)	2.2 ± 1.9	2.2 ± 3.8	0.993
Serum level of FGF21 (pg/mL)	147.4 ± 68.7	227.2 ± 184.0	0.017 *

Abbreviations: FGF, fibroblast growth factor; PTC, papillary thyroid cancer; BMI, body mass index; LDL, low-density lipoprotein; HDL, high-density lipoprotein; AST, aspartate transaminase; ALT, alanine transaminase; TSH, thyroid stimulating hormone; IU, international unit. Data are presented as the mean ± SD. *p*-values from the unpaired *t*-test for continuous parametric variables or the Mann–Whitney U test for nonparametric variables. * *p* < 0.05.

**Table 2 cancers-11-01154-t002:** Clinicopathological parameters of 127 patients according to serum level of FGF21.

Variables		Low FGF21 Group (*n* = 63)	High FGF21 Group (*n* = 64)	*p*-Value
Age (years)	<45	22	20	0.66
	≥45	41	44	
Sex	Male	6	12	0.136
	Female	57	52	
Tumor size (cm)	≤2	57	51	0.088
	>2	6	13	
T stage	T1–T2	33	15	0.001 *
	T3–T4	30	49	
Multicentricity	No	47	46	0.728
	Yes	16	18	
Microscopic capsular invasion	No	27	14	0.011 *
	Yes	36	50	
Extrathyroid extension	No	37	21	0.003 *
	Yes	26	43	
Lymphovascular invasion	No	15	12	0.486
	Yes	48	52	
Central lymph node metastasis	No	42	43	0.95
	Yes	21	21	
Lateral lymph node metastasis	No	57	57	0.793
	Yes	6	6	
Recurrence	No	62	56	0.017 *
	Yes	1	8	
Survival	No	0	4	0.044 *
	Yes	63	60	
BRAFV600E mutation	No	2	1	0.09
	Yes	5	15	
	Unknown	55	48	
Body weight (kg)		48.5 ± 12.1	52.45 ± 12.1	0.068
Height (cm)		158.2 ± 6.7	157.8 ± 9.4	0.773
BMI (kg/m^2^)		22.8 ± 2.8	25.1 ± 3.3	0.013 *
Fasting glucose (mmol/L)		93.5 ± 12.3	99.3 ± 17.5	0.031 *
Triglycerides (mmol/L)		118.7 ± 70.8	153.7 ± 92.1	0.026 *
Total cholesterol (mmol/L)		187.2 ± 40.7	188.7 ± 32.5	0.825
LDL cholesterol (mmol/L)		118.5 ± 35.4	105.5 ± 35.6	0.079
HDL cholesterol (mmol/L)		53.8 ± 12.8	49.2 ± 11.8	0.071
AST (IU/L)		19.5 ± 8.2	19.6 ± 6.2	0.918
ALT (IU/L)		19.1 ± 11.0	21.5 ± 13.7	0.279
TSH (IU/mL)		2.6 ± 5.2	1.9 ± 1.4	0.283

Abbreviations: FGF, fibroblast growth factor; BMI, body mass index; LDL, low-density lipoprotein; HDL, high-density lipoprotein; AST, aspartate transaminase; ALT, alanine transaminase; TSH, thyroid stimulating hormone. T stage was evaluated using the American Joint Committee on Cancer 7th Edition staging system. Data are presented as the mean ± SD. *p*-values are based on unpaired *t*-test for continuous parametric variables and the Mann–Whitney U test for nonparametric variables. The chi-squared test was used for the comparison of categorical variables between groups. * *p* < 0.05.

**Table 3 cancers-11-01154-t003:** Multivariable analysis of the relationships between FGF21 serum levels and clinicopathological factors.

Factors	Exp (β)	SE	95% CI	*p*-Value
T stage (stage 3–4)	3.671	0.776	(0.802, 16.811)	0.094
Microscopic capsular invasion	0.907	0.630	(0.264, 3.120)	0.877
Extrathyroid extension	1.121	0.637	(0.321, 3.908)	0.858
Recurrence	9.985	1.110	(1.135, 87.868)	0.038 *

Abbreviations: FGF, fibroblast growth factor; Exp (β), odds ratio; SE, standard error; CI, confidence interval. Data were analyzed using stepwise logistic regression. * *p* < 0.05.

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
