# Peer review of "Association between Circulating Fibroblast Growth Factor 21 and Aggressiveness in Thyroid Cancer"

_cancers, 2019, doi:10.3390/cancers11081154_

Round 1

Reviewer 1 Report

- The inclusion and exclusion criteria for recruitment of participants in the study should be included in the “Materials and Methods”.

- For Figure 1B, I presume the horizontal lines of the Control and PTC represent the mean serum level of FGF21. It would be desirable to include error bars to indicate 1 SD of the data.

- Figure 1C and lines 110-111. Although the correlation between FGF21 and BMI was significant ( P < 0.05), the correlation coefficient was low (r=0.182) which indicates the correlation was weak. The weak correlation between FGF21 and BMI should be clearly stated in the Results.

- The legend for Figure 2F and 2G were not matched with the two figures. In the legend of Figure 2F, it states the TPC-1 cell migration but the Figure 2F demonstrates the results of BCPAP cell invasion. Similarly, Figure 2G states the BCPAP cell invasion but the Figure 2D demonstrates the result of TPC-1 cell migration.

- The clinical significance of the study findings should be discussed in the Discussion section.

- In the cell viability assay and cell migration and invasion assay, how did the authors determine the concentration of rFGF21?

- Line 321. Please provide information about the clinical diagnosis of the 127 participants leading to the requirement of lobectomy or thyroidectomy.

- Lines 334-335. How was the tumor recurrence of the patients evaluated? By imaging examination such as positron emission tomography? How accurate was the evaluation method? This is important as it affects the reliability of the result for tumour recurrence in Tables 2 and 3.

- Lines 350-351. Please clarify how the diagnosis of thyroid cancer was made? By image-guided fine-needle aspiration cytology?

Author Response

REVIEWER #1

- The inclusion and exclusion criteria for recruitment of participants in the study should be included in the “Materials and Methods”.

A) Thanks for your great comments. We included description of the inclusion and exclusion criteria in the “Materials and Methods” section.

- For Figure 1B, I presume the horizontal lines of the Control and PTC represent the mean serum level of FGF21. It would be desirable to include error bars to indicate 1 SD of the data.

A) Thanks for your comments. We changed bar type included error bars in Figure 1B.

- Figure 1C and lines 110-111. Although the correlation between FGF21 and BMI was significant ( P < 0.05), the correlation coefficient was low (r=0.182) which indicates the correlation was weak. The weak correlation between FGF21 and BMI should be clearly stated in the Results.

A) We agreed with your comments, and indicated the weak correlation between FGF21 and BMI in the “Results” section.

- The legend for Figure 2F and 2G were not matched with the two figures. In the legend of Figure 2F, it states the TPC-1 cell migration but the Figure 2F demonstrates the results of BCPAP cell invasion. Similarly, Figure 2G states the BCPAP cell invasion but the Figure 2D demonstrates the result of TPC-1 cell migration.

A) Thanks for your important comments. We changed descriptions about the indications of Figure 2 in the “Results” section.

- The clinical significance of the study findings should be discussed in the Discussion section.

A) Thank your comments. As you pointed out, we added the clinical significance in ‘‘Discussion’ section.

- In the cell viability assay and cell migration and invasion assay, how did the authors determine the concentration of rFGF21?

A) Thanks for your critical comments. Previous studies about the role of FGF21 in regulation of tissues injury or metabolic disease used the concentration of rFGF21 from 50 ng/mL to 200 ng/ml in vitro (1, 2). Firstly, we hypothesized that the other specific effects of FGF21 existed in tumor cells, we determined the similar concentrations of FGF21 as previous studies. Although there was considerate discrepancy in FGF21 concentration between our preclinical and clinical data, but, as you know, there are many hurdles to represent the human results with in vitro experiments. Additionally, in our knowledge, there were no exact estimates of active forms of circulating FGF21 in blood, since FGF21 was known as the proteolytically processed molecule in the plasma (3-5). A recent advanced study using recombinant FGF21 to treat metabolic disease suggested that physiologically increased serum level of FGF21 in human with the metabolic disease was less than that required to pharmacological action in humans and animals (6, 7). From these results, we carefully suggest that there is the biological role of serum FGF21 in tumor aggressiveness in patients with thyroid cancer. We mentioned the limitation of our study and described the necessities for the further study identifying the active forms of FGF21 in serum to identify the exact biologic process of serum FGF21 in thyroid cancer.

- Line 321. Please provide information about the clinical diagnosis of the 127 participants leading to the requirement of lobectomy or thyroidectomy.

A) Lobectomy were only performed in patients diagnosed as the microcarcinoma (tumor size <25px) without extrathyroidal extension and lymph node metastasis. We mentioned these in the “Materials and Methods” section.

- Lines 334-335. How was the tumor recurrence of the patients evaluated? By imaging examination such as positron emission tomography? How accurate was the evaluation method? This is important as it affects the reliability of the result for tumour recurrence in Tables 2 and 3.

A) In our study, surveillance for recurrent disease usually consisted of physical examinations, measurements of serum levels of thyroglobulin level, anti thyroglobulin levels, ultrasonography every 12 months for 5 years. For patients suspected for disease recurrence, we conducted fine needle aspiration biopsy, whole body scan, and neck CT. The following criteria were used to define recurrence: either pathologic evidence of disease on excision or cytology, or recurrent disease confirmed by elevated Tg and Whole Body Scan. Those patients with no recurrence after a 60-month follow-up evaluated by clinical, laboratorial, and radiological evidence using neck ultrasonography were categorized in nonrecurrent group. The median follow-up period was 84.7 ± 37.4 months to evaluate tumor recurrence. We added these in the “Materials and Methods” section.

- Lines 350-351. Please clarify how the diagnosis of thyroid cancer was made? By image-guided fine-needle aspiration cytology?

A) In our study, all participants were diagnosed as papillary thyroid carcinoma by ultrasound guided fine needle aspiration biopsy. We described the diagnosis of thyroid cancer in the “Materials and Methods” section as your comments.

References

1.         Liu SQ, Roberts D, Kharitonenkov A, Zhang B, Hanson SM, Li YC, et al. Endocrine protection of ischemic myocardium by FGF21 from the liver and adipose tissue. Scientific reports. 2013;3:2767.

2.         Holland WL, Adams AC, Brozinick JT, Bui HH, Miyauchi Y, Kusminski CM, et al. An FGF21-adiponectin-ceramide axis controls energy expenditure and insulin action in mice. Cell metabolism. 2013;17(5):790-7.

3.         Kharitonenkov A, DiMarchi R. FGF21 revolutions: recent advances illuminating FGF21 biology and medicinal properties. Trends in Endocrinology & Metabolism. 2015;26(11):608-17.

4.         Micanovic R, Raches DW, Dunbar JD, Driver DA, Bina HA, Dickinson CD, et al. Different roles of Nand Ctermini in the functional activity of FGF21. Journal of cellular physiology. 2009;219(2):227-34.

5.         Hager T, Spahr C, Xu J, Salimi-Moosavi H, Hall M. Differential enzyme-linked immunosorbent assay and ligand-binding mass spectrometry for analysis of biotransformation of protein therapeutics: application to various FGF21 modalities. Analytical chemistry. 2013;85(5):2731-8.

6.         Coskun T, Bina HA, Schneider MA, Dunbar JD, Hu CC, Chen Y, et al. Fibroblast growth factor 21 corrects obesity in mice. Endocrinology. 2008;149(12):6018-27.

7.         Gaich G, Chien JY, Fu H, Glass LC, Deeg MA, Holland WL, et al. The effects of LY2405319, an FGF21 analog, in obese human subjects with type 2 diabetes. Cell metabolism. 2013;18(3):333-40.

Reviewer 2 Report

The authors set out to determine the potential of FGF21 as a biomarker for aggressiveness in Thyroid cancer. Such a study using human patient samples and cell lines is relevant and can help our understanding of disease pathogenesis and progression. However the study data does not substantiate the claims and needs to be significantly revised.

> Fig 1B: Serum FGF21 levels in patients with PTC were significantly higher than in control participants.”

The authors should magnify the scale of the Y-axis so that the significant difference between the control and PTC patients become obvious. The current image cannot visually substantiate the claim of significant difference.

> Fig 2a. The authors should explain N3. In this sample it appears that’s KLB is upregulated along with FGFR1, FGFR3 and FGFR4. Beta-actin for T4 needs to be repeated.

> Fig 2b. FGF R1 needs to be repeated and perhaps run longer for better resolution of bands.

> Fig 3a. The data provided does not substantiate the claim that FGF21 stimulates downstream signaling through FGFR nor that it triggers EMT.

pFGFR is not upregulated in BCPAP compared to BSA control.

pAKT/ AKT in both BCPAP and TPC-1 is not upregulated compared to control BSA. Additionally, pAKT appears to remain unchanged. Even if the blots are dirty for pAKT antibodies, the upper of the two bands (higher molecular weight due to phosphorylation) usually represent the phosphorylated protein and to show that signaling through the AKT pathway is stimulated, the upper of the two bands should show an increase and also significantly so over the AKT bands.

N-cadherin, E-cadherin and Vimentin does not change relative to the BSA control

> Fig 4 E. The western blot data does not substantiate the claim of rescue of metastasis by FGF21 inhibitor.

GAPDH is not uniform and hence cannot be used as control for the rest of the protein levels. Needs to be repeated.

pAKT shows no change upon treatment with FGF21 inhibitor

Ncadherin, Ecadherin does not show significant change visually at least.  To establish any claim the authors must quantitate the data from 3 repeats of the experiment, using densitometry and then plotting a bar graph showing significant difference between the two groups.

> Fig 4F. The western blot data does not substantiate the claim of rescue of metastasis by FGF21 inhibitor.

AKT and ERK signaling does not seem affected by the addition of FGF21 inhibitor just rehashing that these pathways may not be affected by FGF21 in the first place as seen in the Fig 3a western blots.

N-cadherin and e-cadherin also doesn’t seem affected by inhibitor treatment.

> Since the authors have access to valuable human tissue sample it would be great to see the levels of AKT and ERK pathway proteins.

Author Response

REVIEWER #2

> Fig 1B: “Serum FGF21 levels in patients with PTC were significantly higher than in control participants.” The authors should magnify the scale of the Y-axis so that the significant difference between the control and PTC patients become obvious. The current image cannot visually substantiate the claim of significant difference.

A) Thanks for your great comments. We modified the graph types with SD values and average values as Figure 1B.

> Fig 2a. The authors should explain N3. In this sample it appears that’s KLB is upregulated along with FGFR1, FGFR3 and FGFR4. Beta-actin for T4 needs to be repeated.

A) Thanks for your critical comments. There has been a major mistake in this study, because there was no whole membrane data used in Western blot analysis. We performed all experiments after receiving letters from reviewers, and changed all figures about Western blot analysis. As your comments, we performed analysis using densitometry, to suggest the exact mechanism of FGF21 in tumor aggressiveness. Our revised Fig 2a represented that expression of receptors of FGF21 between normal samples and tumor samples, and we also investigated expression of receptors for FGF21 using The cancer genome atlas. Our data suggested there was no expression of FGF21 in thyroid cancer, but receptors of FGF21 were all expressed in both tumor and normal samples. We described the “Results” section, and “supplementary materials” section.

> Fig 2b. FGFR1 needs to be repeated and perhaps run longer for better resolution of bands.

A) Thanks for your critical comments. We repeated all experiments after receiving letters from reviewers, and changed all figures about Western blot analysis. As your comments, we performed analysis using densitometry, and represented the better resolution of bands.

> Fig 3a. The data provided does not substantiate the claim that FGF21 stimulates downstream signaling through FGFR nor that it triggers EMT. pFGFR is not upregulated in BCPAP compared to BSA control. pAKT/ AKT in both BCPAP and TPC-1 is not upregulated compared to control BSA. Additionally, pAKT appears to remain unchanged. Even if the blots are dirty for pAKT antibodies, the upper of the two bands (higher molecular weight due to phosphorylation) usually represent the phosphorylated protein and to show that signaling through the AKT pathway is stimulated, the upper of the two bands should show an increase and also significantly so over the AKT bands. N-cadherin, E-cadherin and Vimentin does not change relative to the BSA control

A) Thanks for your critical comments. We repeated all experiments after receiving letters from reviewers, and changed all figures about Western blot analysis. As your comments, we performed analysis using densitometry, and represented the better resolution of bands. In revised Fig 3A, the pFGFR, pAKT/AKT, pERK/ERK, and molecules related with EMT signal including N-cadherin, E-cadherin, Vimentin, Slug, and Twist were significantly changed in BCPAP cells with recombinant human FGF21 treatments compared to BSA control.

> Fig 4 E. The western blot data does not substantiate the claim of rescue of metastasis by FGF21 inhibitor. GAPDH is not uniform and hence cannot be used as control for the rest of the protein levels. Needs to be repeated. pAKT shows no change upon treatment with FGF21 inhibitor. Ncadherin, Ecadherin does not show significant change visually at least.  To establish any claim the authors must quantitate the data from 3 repeats of the experiment, using densitometry and then plotting a bar graph showing significant difference between the two groups. Fig 4F. The western blot data does not substantiate the claim of rescue of metastasis by FGF21 inhibitor. AKT and ERK signaling does not seem affected by the addition of FGF21 inhibitor just rehashing that these pathways may not be affected by FGF21 in the first place as seen in the Fig 3a western blots. N-cadherin and e-cadherin also doesn’t seem affected by inhibitor treatment.

A) As your comments, we performed analysis using densitometry, and represented the better resolution of bands. In revised Fig 4 and supplementary figure 7, the pFGFR, pAKT/AKT, pERK/ERK, and molecules related with EMT signal were significantly rescued by FGFR inhibitor. We investigated about expression of Twist and Slug to identify the regulation of FGF21 on EMT signal, and FGFR inhibitor treatment inhibited EMT signaling axis induced by rhFGF21 treatment. These results could support our hypothesis that the FGF21 have a significant role in tumor aggressiveness in thyroid cancer.

> The inclusion and exclusion criteria for recruitment of participants in the study should be included in the “Materials and Methods”.

A) We describe the inclusion and exclusion criteria for recruitment of participants in the study should be included in the “Materials and Methods” as your comments.

Reviewer 3 Report

This is an interesting paper despite the several limitations. 1. I also would like to see the univariate analysis for clinicopathologic features 2. There almost were an significant differences in BRAF mutations occurence, is there any relationship between those? 3. Impact on FGF21 level on clinical predisctors such as OS, DFS etc. could be interesting

Author Response

> I also would like to see the univariate analysis for clinicopathologic features

A) Thanks about your great comments. We investigated the correlation between serum FGF21 levels and clinicopathlogic features as below. We added about these in the section of “results” and “supplementary materials”.

Correlation analysis between serum FGF21 level and metabolic parameters in total participants with PTC (N = 127)

Variables

Mean   ± SD or number of patients (%)

Age

0.131

0.092

BMI   (kg/m2)

0.182

0.040*

Fasting   glucose (mg/dL)

0.015

0.850

Triglycerides   (mg/dL)

0.223

0.006**

Total   cholesterol (mg/dL)

0.125

0.107

LDL   cholesterol (mg/dL)

0.018

0.840

HDL   cholesterol (mg/dL)

-0.140

0.017*

AST   (IU/L)

0.053

0.498

ALT   (IU/L)

0.042

0.590

Data are presented as means ± SDs or number of patients (%)

Abbreviation: BMI, body mass index. LDL, low density lipoprotein. HDL, high density lipoprotein. AST, aspartate transaminase, ALT, alanine transaminase

a Coefficients (r) were calculated using Spearman’s method. * p-value <0.05,** p-value <0.01.

> There almost were an significant differences in BRAF mutations occurence, is there any relationship between those?

A) Thanks for your wonderful comments. Since presence of BRAF mutation, well known oncogene related tumor aggressiveness, was able to influence tumor behavior, we checked the presence of BRAF mutation. Unfortunately, most of participants in this study, was diagnosed as the papillary thyroid cancer before 8 years ago, and there was no data about the presence of BRAF mutation. Since only 23 patients were evaluated by detection of BRAFV600E mutation on fine needle aspiration specimens of thyroid before surgery, our study could not exactly validate the relationship between FGF21 and BRAF mutation’s occurrence. However, as shown in table 2, there was no significance between BRAF mutation and serum FGF21 levels (P=0.090). We also investigated the relationship between FGF21 and BRAF mutation in 23 patients evaluated the presence of BRAF mutation, and there was also no significant difference (P=0.144). As we mentioned in discussion section, the further large cohort study was necessary to clarify the role of FGF21 in thyroid cancer.

FGF21   low group

FGF 21   high group

BRAFV600E mutation

No

Yes

Unknown

2

5

55

1

15

48

0.090

BRAFV600E mutation

No

Yes

2

5

1

15

0.144

The Chi-square test was used for the comparison of the categorical variables between groups.

> Impact on FGF21 level on clinical predictors such as OS, DFS etc. could be interesting

A) Thank you for your good opinions about our study. We calculated the relationship of OS, RFS with serum FGF21 levels. There were significant association of FGF21 and survival graphs, and we added these results in “Results” sections.

Round 2

Reviewer 2 Report

1. REVIEWER 2 COMMENT  “The authors should explain N3. In this sample it appears that’s KLB is upregulated along with FGFR1, FGFR3 and FGFR4. Beta-actin for T4 needs to be repeated”

The authors state that they have repeated the experiment with whole membrane but level of KLB still remains the same. The authors should explain the data or hypothesize what could be going on with that sample or use another sample to generalize any conclusion about KLB from this experiment.

2. AUTHOR REVISION COMMENT "In revised Fig 3A, the pFGFR, pAKT/AKT, pERK/ERK, and molecules related with EMT signal including N-cadherin, E-cadherin, Vimentin, Slug, and Twist were significantly changed in BCPAP cells with recombinant human FGF21 treatments compared to BSA control."

Revised Fig 3A does not show the above data. The authors need to be careful while labellig and discussing data. The data mentioned above is shown in Fig 4A. The change in level of pERK CANNOT be considered significant. The authors should describe how significance was calculated for densitometry and explain the discrepancy in the blot vs the densitometry plots. The densitometry should only substantiate the blots and should not be opposed to it. N-cadherin levels in the blots clearly DON’T CHANGE but the graph shows a significant increase! The authors should explain how that is possible. By extension, no conclusion from this figure is acceptable until the scientific basis of the generation of the figure is clarified.  

2. REVISED FIGURE "Figure 4. Effects of FGF21 in the FGFR signaling axis, including epithelialmesenchymal transition-associated protein in thyroid cancer cell lines. (A) Representative images of Western blot analyses for the detection of p-FGFR, p-AKT, AKT, p-ERK, ERK, N-cadherin, E-cadherin, VIMENTIN, SLUG, TWIST, and Beta-actin in BCPAP cells treated with rFGF21, or 0.1% BSA (B) Representative images of Western blot analyses for the detection of p-FGFR, p-AKT, AKT, p-ERK, ERK, N-cadherin, E-cadherin, VIMENTIN, SLUG, TWIST, and Beta-actin in BCPAP cells treated with rFGF21, or 0.1%. Experiments were performed in duplicate, and all experiments were performed at least three times. **means P<0.01, * means P<0.05 compared to 0.1% BSA control. "

4a. Legend is a duplication of 4B. Hence no conclusion can be drawn from the data provided unless it is clear what they actually are. The authors need to be more careful and cognizant about the revision they submit in future for it to be considered for publication.  

Author Response

Thanks for your great comments. Revised manuscript had undergone English editing as your comments, and we did great efforts to proof your questions. Very thank you for your advisor.
